# The role of stress and health behaviour in linking weight discrimination and health: a secondary data analysis in England

Ruth A Hackett ![ORCID],[1] Sarah E Jackson ![ORCID],[2] Elizabeth Corker,[3,4] Andrew Steptoe ![ORCID] [2]

[1]Health Psychology Section, Department of Psychology, King's College London, London, UK
[2]Department of Behavioural Science and Health, University College London, London, UK
[3]Clinical, Educational and Health Psychology, University College London, London, UK
[4]Department of Psychology, University of Sheffield, Sheffield, UK

**Correspondence to**
Dr Ruth A Hackett;
ruth.hackett@kcl.ac.uk

## ABSTRACT

**Objective** To examine the role of stress and health-risk behaviours in relationships between weight discrimination and health and well-being.

**Design** Secondary data analysis of an observational cohort study.

**Setting** The English Longitudinal Study of Ageing.

**Participants** Data were from 4341 adults (≥50 years) with overweight/obesity.

**Primary outcome measures** We tested associations between perceived weight discrimination at baseline (2010/2011) and self-rated health, limiting long-standing illness, depressive symptoms, quality of life and life satisfaction over 4-year follow-up (2010/2011; 2014/2015). Potential mediation by stress exposure (hair cortisol) and health-risk behaviours (smoking, physical inactivity, alcohol consumption) was assessed.

**Results** Cross-sectionally, perceived weight discrimination was associated with higher odds of fair/poor self-rated health (OR=2.05 (95% CI 1.49 to 2.82)), limiting long-standing illness (OR=1.76 (95% CI 1.29 to 2.41)) and depressive symptoms (OR=2.01 (95% CI 1.41 to 2.85)) and lower quality of life (B=−5.82 (95% CI −7.01 to −4.62)) and life satisfaction (B=−2.36 (95% CI −3.25 to −1.47)). Prospectively, weight discrimination was associated with higher odds of fair/poor self-rated health (OR=1.63 (95% CI 1.10 to 2.40)) and depressive symptoms (OR=2.37 (95% CI 1.57 to 3.60)) adjusting for baseline status. Those who reported discrimination had higher hair cortisol concentrations (B=0.14 (95% CI 0.03 to 0.25)) and higher odds of physical inactivity (OR=1.90 (95% CI 1.18 to 3.05)). These variables did not significantly mediate associations between discrimination and health outcomes.

**Conclusions** Weight discrimination is associated with poor health and well-being. While this discrimination is associated with stress exposure and physical inactivity, these variables explain little of the association between discrimination and poorer outcomes.

## STRENGTHS AND LIMITATIONS OF THIS STUDY

⇒ Secondary data analysis of a large prospective observational study.
⇒ Assessment of multiple health and well-being outcomes.
⇒ The prevalence of weight discrimination in the sample was low (which limits generalisability) and it was only assessed on one occasion increasing the risk of misclassification bias.
⇒ Measures of health behaviour were self-reported increasing the risk of measurement error and there was no data on diet (patterns or quality) or eating behaviour (eg, dieting, disordered eating).
⇒ The sample comprised of mostly white middle aged and older adults which limits generalisability.

(25–29.9 kg/m$^2$) or obesity (≥30 kg/m$^2$) range, and a shift in the distribution of cases such that a greater proportion have 'morbid' obesity (BMI≥40 kg/m$^2$).[3] Despite the increasingly normative nature of obesity, it is widely stigmatised with those with obesity characterised as lazy, unintelligent and lacking in self-discipline.[4] These stereotypes translate into weight discrimination across contexts and interactions, including the workplace, education, healthcare settings and interpersonal relationships.[4]

Mounting evidence suggests that weight discrimination can impact health and well-being.[5 6] Studies have linked perceived weight discrimination to adverse outcomes, including poorer self-rated health, greater disease burden, functional disability, psychiatric comorbidity, psychosocial stress, loneliness, poorer life satisfaction and increased risk of premature death.[7–13] Evidence is suggestive of a prospective relationship (ie, discrimination preceding a decline in health) which persists after adjustment for BMI.[7 9] It is of interest to analyse these issues in middle-aged

## INTRODUCTION

The average person in the UK carries excess weight.[1 2] Recent decades have seen an increase in the proportion of adults with a body mass index (BMI) in the overweight

and older people, among whom disease and disability risk is elevated.

Several mechanisms may underpin the relationship between perceived weight discrimination and poorer health and well-being. Prominent theories focus on the role of (1) stress responses to discrimination and (2) health-risk behaviours as a means of coping with or avoiding discrimination.[14] Experiencing weight discrimination can be stressful, both physiologically, in the form of cortisol responses[15 16] and psychologically, in the form of increased negative emotion (eg, psychological distress, depressive symptoms and fear, among others).[8] If an individual perceives discrimination regularly, as is common with weight stigma,[17] stress responses will be activated frequently, potentially leading to chronically elevated cortisol and a negative emotional state (eg, depression, anxiety, ongoing feelings of stress). This in turn can lead to ill health under allostatic load theory.[18] Indeed, observational studies have associated elevated cortisol levels and weight discrimination.[19 20]

Health-risk behaviours may also link weight discrimination to poorer health and well-being. These may emerge as a biological response to discrimination, for example, hypothalamic-pituitary-adrenal (HPA) axis activation upregulating appetite and the drive to consume energy-dense 'comfort' foods.[21–23] Indeed, disordered eating and binge eating disorder (eating large amounts accompanied by feelings of distress) is common at higher weight ranges[24] and has been associated with weight discrimination.[25] In this way, weight stigma may contribute to a vicious cycle of stress, weight gain and further discrimination.[14] Unhealthy behaviours may also provide a coping mechanism: food, nicotine and alcohol activate dopaminergic reward pathways in the brain[26] and may help to alleviate the negative psychological impact of discrimination. Avoiding activities for fear of discrimination (eg, exercising in public places) may also act as a barrier to a healthy lifestyle.[27] To our knowledge, no studies have investigated the extent to which these variables mediate associations between weight-based discrimination and health and well-being outcomes.

The present study used data from the English Longitudinal Study of Ageing (ELSA) to examine (1) cross-sectional and prospective associations between perceived weight discrimination and health and well-being (self-rated health, limiting long-standing illness, depressive symptoms, quality of life and life satisfaction) and (2) whether these associations were mediated by stress exposure and health-risk behaviours (smoking, physical inactivity and alcohol consumption).

## METHODS
### Study population
ELSA is a representative longitudinal study of adults aged ≥50 living in England.[28] Established in 2002, ELSA collects data via computer-assisted personal interviews and self-completion questionnaires biennially, with nurse visits in alternate waves to collect objective health measures, including BMI.

We used data on perceived weight discrimination from Wave 5 (2010/2011; the only wave in which discrimination was assessed); hair cortisol, health behaviours and BMI from Wave 6 (2012/2013), and health and well-being from Waves 5 and 7 (2014/2015). We restricted our sample to those with a BMI in the overweight or obese range (≥25 kg/m$^2$).

Data are freely available to download from https://g2aging.org.[29]

### Patient and public involvement
No patient involved.

### Measures
All questionnaire measures used in the study can be found in online supplemental material.

#### Exposure: perceived weight discrimination
Our measure of perceived discrimination was based on items developed and used in other prospective cohort studies (eg, Health and Retirement Study; Midlife Development in the USA).[30] Participants were asked how often they encounter five discriminatory situations: 'In your day-to-day life, how often have any of the following things happened to you: (1) you are treated with less respect or courtesy; (2) you receive poorer service than other people in restaurants and stores; (3) people act as if they think you are not clever; (4) you are threatened or harassed; and (5) you receive poorer service or treatment than other people from doctors or hospitals'. Response options were on a 6-point scale ranging from 'never' to 'almost every day'. Because data were skewed, with most participants reporting no discrimination, we dichotomised responses to indicate whether or not participants had experienced discrimination in the past year (a few times or more a year vs less than once a year or never), with the exception of the fifth item which was dichotomised to indicate whether or not respondents had ever experienced discrimination from doctors or hospitals (never vs all other options) as most participants never reported discrimination in this setting.[31–34]

A follow-up question asked participants who reported discrimination to indicate the reason(s) they attributed to their experience (eg, weight, age, sex, race, sexual orientation, disability).[31–34] Those who attributed any experience of discrimination to their weight are treated as cases of perceived weight discrimination as in other studies.[19 35–39]

#### Outcomes: health and well-being
Self-rated health was assessed using a single item: 'Would you say your health is… poor/fair/good/very good/excellent?' We analysed the proportion of individuals rating their health as fair/poor, as in other investigations.[31 40] As a sensitivity check, analyses were also conducted using the continuous range of scores.

Limiting long-standing illness was assessed with two questions: (1) 'Do you have any long-standing illness, disability, or infirmity? By long-standing I mean anything that has troubled you over a period of time or that is likely to affect you over a period of time'. Those who responded yes were asked: (2) 'Does this illness or disability limit your activities in any way?' Those who responded yes to both items were classed as having a limiting long-standing illness.

Depressive symptoms were assessed with the eight-item Center for Epidemiologic Studies Depression Scale.[41] Respondents were asked if they had experienced depressive symptoms (eg, restless sleep and being unhappy) over the past month using a binary (yes/no) response. Total scores ranged from 0 to 8. Data were dichotomised using an established cut-off (scores ≥4).[42] As a sensitivity check, analyses were also conducted using the continuous range of scores.

Quality of life was assessed with the Control, Autonomy, Self-realisation and Pleasure-19 (CASP-19) scale .[43] Items covered four domains of quality of life; control (eg, 'I feel that what happens to me is out of my control'), autonomy (eg, 'My health stops me from doing things I want to do'), self-realisation (eg, 'I feel that life is full of opportunities') and pleasure (eg, 'I enjoy being in the company of others'). Respondents were asked how often each statement applies to them (often=0, sometimes=1, not often=2, never=3). Positively-worded items were reverse scored so that a higher total score indicated higher quality of life (range: 0–57).

Life satisfaction was assessed with the 5-item Satisfaction With Life Scale,[44] which asks the extent to which participants agree with statements including 'In most ways my life is close to my ideal'; and 'I am satisfied with my life'. Responses ranged from 0 (strongly disagree) to 6 (strongly agree) and were summed to produce a total score of 0–30.

### Mediators: stress exposure and health-risk behaviours

The hair cortisol assessment procedure has been described in detail elsewhere.[45] Briefly, a scalp-proximal hair sample at least 2 cm long and weighing at least 10 mg was taken from the posterior vertex, cut as close to the scalp as possible. The wash procedure and steroid extraction were undertaken using high performance liquid chromatography–mass spectrometry. Assuming an average hair growth of 1 cm per month the 2 cm hair segment represents average cortisol accumulated 2 months prior to sampling.

Smoking status was assessed with the question 'Do you smoke cigarettes at all nowadays?' (yes/no).

Physical activity was self-reported in response to three questions on the frequency of participation in light, moderate and vigorous activities (more than once a week/once a week/one to three times a month/hardly ever or never). We compared participants who were physically inactive (defined as no activity on a weekly basis)

with those who reported engaging in light, moderate or vigorous activity at least once a week.

Alcohol consumption was assessed with the question 'On how many days out of the last seven did you have an alcoholic drink?'. Those who responded 'five', 'six' or 'seven' days were classified as frequent alcohol drinkers. We compared frequent alcohol drinkers with all other categories combined.

### Covariates

Participants self-reported their age, sex and ethnicity (white, other). Socioeconomic status was indexed with a measure of household non-pension wealth[28], divided into quintiles across the entire Wave 5 sample. BMI $(kg/m^2)$ was based on objective height and weight measurements.

### Statistical analysis

Associations between perceived weight discrimination and covariates were examined using one-way independent analysis of variance for continuous variables and $\chi^2$ tests for categorical variables.

For our primary analyses, we used multivariable regression models (logistic for categorical outcomes, linear for continuous outcomes) to explore cross-sectional and prospective associations between perceived weight discrimination and health and well-being. All models were adjusted for age, sex, ethnicity, wealth and BMI. Prospective models were additionally adjusted for baseline status/score on the outcome variable.

Where perceived weight discrimination was significantly associated with an outcome in prospective analyses, we tested for mediation by stress exposure and health-risk behaviours (figure 1). Hair cortisol data was log transformed to correct skewness. Establishing mediation requires the mediator to be associated with the exposure (path a) and the outcome (path b), so we first tested associations of (1) perceived weight discrimination with (a) stress exposure (hair cortisol concentration) using linear regression and (b) health-risk behaviours (smoking, physical inactivity and alcohol consumption) using logistic regression; and (2) stress exposure and health-risk behaviours with health and well-being outcomes using logistic or linear regression, as appropriate. Where associations were significant, we used the *sgmediation* command in Stata, which calculates total (path c), direct (path c') and indirect (path a×b) effects, and tests the significance of the indirect effect using the Sobel test. We used bootstrapping with 5000 sampling replications to estimate the 95% CI and calculated effect ratios reflecting the proportion of the total effect of the independent variable on the dependent variable that is explained by the mediator. Mediation models were adjusted for age, sex, ethnicity, wealth, BMI and status/score on the outcome variable at baseline.

We conducted three sensitivity analyses. In the first we expanded our analytical sample to include participants who provided data on perceived weight discrimination but did not have a BMI≥25 $kg/m^2$. We assessed whether

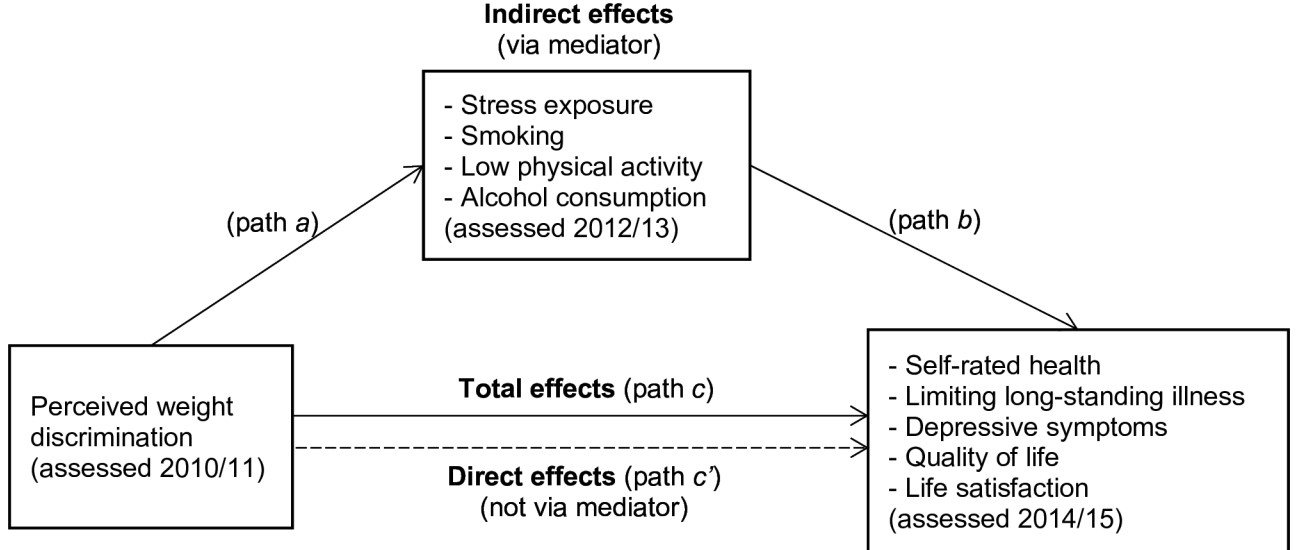

**Figure 1** Mediation model of associations between perceived weight discrimination and health and well-being outcomes via health behaviours and stress exposure.

the pattern of associations between perceived weight discrimination and health and well-being outcomes was similar when including these participants. For our second sensitivity analysis we restricted our analytical sample to participants with BMI≥30 kg/m² and assessed whether the associations between perceived weight discrimination and health and well-being outcomes remained for those participants in the higher BMI ranges. For the final sensitivity analysis, we changed the frequency weight discrimination was encountered from 'in the past year vs less than once a year/never' (main analysis) to whether the participant *ever* experienced discrimination ('never' vs 'almost every day; at least once a week; a few times a month; a few times a year; less than once a year'). We assessed whether our results changed when using this different definition of weight discrimination. Analyses were performed on complete cases using SPSS V.25, with the exception of the mediation models which were run in Stata V.13.

## RESULTS

In Wave 5, 9090 participants (77.5% of those eligible) took part in the face-to-face interview, of whom 8107 (93%) returned the self-completion questionnaire. We excluded 2142 participants with missing data on weight discrimination or covariates (including BMI) and 1624 with BMI<25 kg/m². Our final sample for cross-sectional analysis comprised of 4341 people, aged 52–89 years with a BMI 25.0–59.4 kg/m².

Perceived weight discrimination was reported by 210 (4.8%) respondents, who were on average younger, heavier, more socioeconomically disadvantaged and more likely to be women than those who did not report weight discrimination (table 1). Of the participants who perceived weight discrimination, 44 (21%) also reported sex discrimination, 40 (19%) reported disability

discrimination, 11 (5.2%) reported race discrimination and 6 (2.9%) reported sexuality discrimination.

Cross-sectionally, perceived weight discrimination was associated with significantly higher odds of fair/poor self-rated health, limiting long-standing illness and depressive symptoms, and lower mean ratings of quality of life and life satisfaction, after adjustment for age, sex, ethnicity, wealth and BMI (table 2). Prospectively, perceived weight discrimination was associated with significantly higher odds of future fair/poor self-rated health and depressive symptoms, but differences in limiting long-standing illness, quality of life and life satisfaction were not statistically significant after adjustment for covariates including baseline status/score on the outcome variable (table 2). Analysing depression and self-rated health as continuous variables did not change the pattern of results (table 2).

We assessed whether prospective associations observed between perceived weight discrimination and self-rated health and depressive symptoms were mediated by stress exposure and health-risk behaviours (figure 1; as limiting long-standing illness, quality of life and life satisfaction were not prospectively associated with discrimination these were not investigated). After adjustment for covariates, participants who reported weight discrimination in Wave 5 had significantly higher hair cortisol concentrations (indicative of stress exposure) and higher odds of physical inactivity in Wave 6 than those who did not (table 3; path *a* figure 1). No association with smoking or alcohol consumption was observed (table 3). Physical inactivity in Wave 6 was significantly associated with higher odds of fair/poor self-rated health and depressive symptoms in Wave 7 (table 4; path *b* figure 1). While hair cortisol concentrations in Wave 6 were not significantly associated with depressive symptoms in Wave 7 (table 4; path *b* figure 1), each one-unit increase in hair cortisol concentration (log pg/mg) was associated with 17%

**Table 1** Sample characteristics at baseline (Wave 5)

| | Whole sample (n=4341) | Perceived weight discrimination (n=210) | No perceived weight discrimination (n=4131) | P value |
|---|---|---|---|---|
| Age (years), mean (SD) | 66.62 (7.96) | 62.04 (6.58) | 66.85 (7.95) | <0.001 |
| Sex, % (n) | | | | |
| Men | 47.2 (2051) | 34.8 (73) | 47.9 (1978) | <0.001 |
| Women | 52.8 (2290) | 65.2 (137) | 52.1 (2153) | – |
| Ethnicity, % (n) | | | | |
| White | 97.9 (4250) | 97.6 (205) | 97.9 (4045) | 0.768 |
| Other | 2.1 (91) | 2.4 (5) | 2.1 (86) | – |
| Wealth quintile, % (n) | | | | |
| 1 (poorest) | 15.5 (672) | 26.7 (56) | 14.9 (616) | <0.001 |
| 2 | 20.6 (896) | 29.0 (61) | 20.2 (835) | – |
| 3 | 21.0 (913) | 18.6 (39) | 21.2 (874) | – |
| 4 | 21.8 (946) | 16.2 (34) | 22.1 (912) | – |
| 5 (richest) | 21.1 (914) | 9.5 (20) | 21.6 (894) | – |
| BMI (kg/m$^2$), mean (SD) | 30.37 (4.47) | 36.63 (6.17) | 30.05 (4.12) | <0.001 |

BMI, body mass index.

higher odds of subsequently reporting fair/poor self-rated health, which while not quite statistically significant (p=0.066) warranted further investigation as a potential mediator.

Results of the mediation analyses are summarised in table 5 (path *c*, path *c'* and indirect effects figure 1). There was no evidence of mediation, with effect ratios indicating that differences in physical inactivity explained approximately 4.2% of the association between perceived weight discrimination and self-rated health and 1.5% of the association between perceived weight discrimination and depressive symptoms, and differences in hair cortisol concentrations explained 1.5% of the association between perceived weight discrimination and self-rated health. None of these indirect effects were statistically significant.

### Sensitivity analyses

In the first sensitivity analysis, we expanded our sample to include participants who provided data on perceived weight discrimination but did not have a BMI in the overweight/obesity range (see online supplemental table 1 for sample characteristics). We assessed whether the pattern of associations between weight discrimination and health and well-being outcomes was similar when including these participants (online supplemental table 2). The results remained unchanged when including participants who did not have overweight/obesity.

In the second sensitivity analysis, we restricted the sample to participants with BMI≥30 kg/m$^2$ (see online supplemental table 3 for the characteristics of this restricted sample). We assessed whether the observed relationships between perceived weight discrimination and health and well-being outcomes were similar in these participants in the higher BMI ranges. As can be seen the

results were similar when restricting the sample to those with ≥30 kg/m$^2$ (see online supplemental table 4).

For our final sensitivity analysis, we changed the frequency weight discrimination was encountered from *at least once a year* (main analysis) to whether the participant *ever* experienced discrimination ('never' vs all other options). Ever encountering weight discrimination had a similar impact on cross-sectional and prospective health and well-being outcomes (online supplemental table 5).

### DISCUSSION

In this large cohort of middle-aged and older adults, cross-sectionally, perceived weight discrimination was associated with higher odds of fair/poor self-rated health, limiting long-standing illness and depressive symptoms, and lower quality of life and life satisfaction. Prospectively, perceived weight discrimination was associated with higher odds of fair/poor self-rated health and depressive symptoms after adjustment for baseline status. All associations were independent of age, sex, ethnicity, wealth and BMI. However, while those who reported weight discrimination had higher odds of physical inactivity and higher hair cortisol concentrations than those who did not, there was no evidence that these variables mediated associations between perceived weight discrimination and poor health and well-being.

Our finding that perceived weight discrimination is associated with poorer health and well-being is consistent with previous studies.[7–11] While our results provide prospective evidence linking perceived weight discrimination with self-rated health and depressive symptoms, as has been observed in other samples,[7 8 11] we failed to replicate a prospective association with life satisfaction[7]

**Table 2** Cross-sectional and prospective associations of perceived weight discrimination with health and well-being outcomes (path *c* on figure 1)

| | Cross-sectional (Wave 5) | | Prospective (Wave 7) | |
|---|---|---|---|---|
| | No perceived weight discrimination | Perceived weight discrimination | No perceived weight discrimination | Perceived weight discrimination |
| **Fair/poor self-rated health** | | | | |
| n included in analysis | 4130 | 208 | 3705 | 188 |
| % (n) reporting outcome | 22.5 (928) | 48.1 (100) | 26.3 (973) | 53.7 (101) |
| Adjusted OR (95% CI) | 1.00 (ref) | 2.05 (1.49 to 2.82) | 1.00 (ref) | 1.63 (1.10 to 2.40) |
| P value | | <0.001 | | 0.014 |
| **Self-rated health (range 1–5)** | | | | |
| n included in analysis | 4130 | 208 | 3705 | 188 |
| Mean (SD) score | 2.73 (1.04) | 3.47 (1.06) | 2.85 (1.05) | 3.56 (0.99) |
| Adjusted B (95% CI) | Ref | 0.44 (0.29 to 0.58) | Ref | 0.14 (0.02 to 0.27) |
| P value | | <0.001 | | 0.026 |
| **Limiting long-standing illness** | | | | |
| n included in analysis | 4129 | 209 | 3740 | 190 |
| % (n) reporting outcome | 32.2 (1330) | 55.0 (115) | 36.2 (1354) | 56.8 (108) |
| Adjusted OR (95% CI) | 1.00 (ref) | 1.76 (1.29 to 2.41) | 1.00 (ref) | 1.24 (0.85 to 1.79) |
| P value | | <0.001 | | 0.268 |
| **Depressive symptoms** | | | | |
| n included in analysis | 4098 | 210 | 3643 | 188 |
| % (n) reporting outcome | 11.9 (487) | 28.6 (60) | 10.8 (392) | 29.8 (56) |
| Adjusted OR (95% CI) | 1.00 (ref) | 2.01 (1.41 to 2.85) | 1.00 (ref) | 2.37 (1.57 to 3.60) |
| P value | | <0.001 | | <0.001 |
| **Depressive symptoms (range 0–8)** | | | | |
| n included in analysis | 4098 | 210 | 3643 | 188 |
| Mean (SD) score | 1.30 (1.78) | 2.60 (2.42) | 1.24 (1.73) | 2.31 (2.12) |
| Adjusted B (95% CI) | Ref | 0.94 (0.68 to 1.20) | Ref | 0.38 (0.15 to 0.62) |
| P value | | <0.001 | | <0.001 |
| **Quality of life (range 0–57)** | | | | |
| n included in analysis | 3942 | 203 | 3106 | 154 |
| Mean (SD) score | 41.72 (8.36) | 34.24 (9.57) | 42.19 (8.47) | 36.56 (9.34) |
| Adjusted B (95% CI) | Ref | -5.82 (–7.01 to –4.62) | Ref | 0.01 (–0.96 to 0.97) |
| P value | | <0.001 | | 0.992 |
| **Life satisfaction (range 0–30)** | | | | |
| n included in analysis | 4008 | 199 | 3240 | 160 |
| Mean (SD) score | 21.05 (6.02) | 17.45 (7.62) | 21.22 (6.03) | 18.46 (7.31) |
| Adjusted B (95% CI) | Ref | -2.36 (–3.25 to –1.47) | Ref | -0.15 (–0.89 to 0.59) |
| P value | | <0.001 | | 0.691 |

ORs, Bs and 95% CIs are adjusted for age, sex, ethnicity, wealth and body mass index. Prospective results are additionally adjusted for status/score on the outcome variable at baseline (Wave 5).
ref, reference category.

or detect any prospective association with limiting long-standing illness or quality of life. It is unclear whether this is the result of differences in measurement (eg, assessing limiting long-standing illness vs total disease burden), sample (eg, age range, the lack of ethnic diversity in the current sample), setting (ie, England vs the USA) or covariates used (highlighting the possibility of uncontrolled confounding). Nonetheless, taking our findings in the context of the literature, it is evident that weight stigma is a social determinant of health and a driver of health inequalities.[46] As such, it is not only a social justice issue, but also a public health issue. Further, we

**Table 3** Associations of perceived weight discrimination at Wave 5 with health behaviours and stress exposure at Wave 6 (path *a* on figure 1)

| | No perceived weight discrimination | Perceived weight discrimination |
|---|---|---|
| Smoking status | | |
| n included in analysis | 4131 | 210 |
| % (n) current smoker | 8.4 (347) | 14.3 (30) |
| Adjusted OR (95% CI) | 1.00 (ref) | 1.43 (0.91 to -2.25) |
| P value | | 0.120 |
| Physical inactivity | | |
| n included in analysis | 4131 | 210 |
| % (n) physically inactive | 5.9 (242) | 13.8 (29) |
| Adjusted OR (95% CI) | 1.00 (ref) | 1.90 (1.18 to 3.05) |
| P value | | 0.008 |
| Alcohol consumption | | |
| n included in analysis | 3890 | 186 |
| % (n) frequent alcohol consumption | 20.6 (801) | 12.9 (24) |
| Adjusted OR (95% CI) | 1.00 (ref) | 0.93 (0.59 to 1.49) |
| P value | | 0.774 |
| Hair cortisol concentration | | |
| n included in analysis | 2456 | 128 |
| Mean (SD) log pg/mg | 0.90 (0.57) | 1.09 (0.65) |
| Adjusted B (95% CI) | Ref | 0.14 (0.03 to 0.25) |
| P value | | 0.012 |

ORs and 95% CIs are adjusted for age, sex, ethnicity, wealth and body mass index.
ref, reference category.

observed an inverse association between perceived weight discrimination and wealth, suggesting that those of lower socioeconomic position could be more vulnerable to the deleterious effects of weight discrimination. This inverse association is in line with previous reports in ELSA, but not other studies.[47] Therefore, the role of socioeconomic position in the associations between weight discrimination, health and well-being, warrants further attention.

A key study aim was to explore the role of stress and health-risk behaviours as pathways underpinning associations between weight discrimination and health and well-being.[48] In line with previous findings, perceived weight discrimination was associated with increased stress (higher hair cortisol concentrations) and physical inactivity,[19 20] but these variables explained only a very small and non-significant proportion of the total associations of perceived weight discrimination with self-rated health and depressive symptoms. Associations with smoking and alcohol consumption were not statistically significant, so these variables were not considered plausible mediators. There are several potential explanations for these findings. The variables we assessed make only small contributions to the impact of weight discrimination on health and well-being, and there are other variables more important to this relationship that we were unable to account for in our modelling. We had no information on dietary factors or eating behaviours such as unsupervised and/or unsafe dieting practices, binge eating, disordered eating and weight cycling. Therefore, we were unable to explore these as potential mediators. Given evidence that people who experience weight discrimination are at increased risk of overeating, disordered eating[49 50] and have a greater risk of binge eating disorder[25] than those who do not report discrimination these present important unexamined pathways from weight discrimination to poor health. It is plausible that the use of maladaptive dietary and eating behaviours as a means of coping with weight discrimination could lead to a vicious cycle of further distress, weight gain and disproportionate burden of weight discrimination.[14 25] This represents an important avenue of further research, particularly considering the lack of mediation by other factors in the current study. Future prospective studies should explicitly set out to measure the role of

**Table 4** Associations of level of physical activity and stress exposure at Wave 6 with health and well-being outcomes (prospectively associated with perceived weight discrimination) at Wave 7 (path *b* on figure 1)

| | Fair/poor self-rated health | | Depressive symptoms | |
|---|---|---|---|---|
| | No | Yes | No | Yes |
| Physical inactivity | | | | |
| n included in analysis | 2821 | 1075 | 3404 | 456 |
| % (n) physically inactive | 3.4 (96) | 12.4 (133) | 5.1 (173) | 11.4 (52) |
| Adjusted OR (95% CI) | 1.00 (ref) | 2.57 (1.92 to 3.45) | 1.00 (ref) | 1.71 (1.21 to 2.42) |
| P value | | <0.001 | | 0.002 |
| Hair cortisol concentration | | | | |
| n included in analysis | 1699 | 623 | 2008 | 292 |
| Mean (SD) log pg/mg | 0.89 (0.57) | 0.97 (0.58) | 0.91 (0.57) | 0.91 (0.57) |
| Adjusted OR (95% CI) | 1.00 (ref) | 1.17 (0.99 to 1.38) | 1.00 (ref) | 0.93 (0.74 to 1.17) |
| P value | | 0.066 | | 0.538 |

ORs and 95% CIs are adjusted for age, sex, ethnicity, wealth and body mass index.
ref, reference category.

Table 5  Models testing mediation of the associations between perceived weight discrimination and fair/poor self-rated health and depressive symptoms by level of physical activity and stress exposure (see figure 1)

| | Mediation by physical inactivity | | | | |
| --- | --- | --- | --- | --- | --- |
| | Coeff. | SE | P value | Bootstrap 95% CI | Effect ratio |
| Fair/poor self-rated health | | | | | |
| Total effect (path *c*) | 0.081 | 0.030 | 0.007 | – | – |
| Direct effect (path *c'*) | 0.077 | 0.030 | 0.009 | – | – |
| Indirect effect (via mediator) | 0.003 | 0.002 | 0.062 | -0.0002 to 0.009 | 0.042 |
| Depressive symptoms | | | | | |
| Total effect (path *c*) | 0.113 | 0.023 | <0.001 | – | – |
| Direct effect (path *c'*) | 0.111 | 0.023 | <0.001 | – | – |
| Indirect effect (via mediator) | 0.002 | 0.001 | 0.169 | -0.001 to 0.005 | 0.015 |
| | Mediation by hair cortisol concentration | | | | |
| Fair/poor self-rated health | | | | | |
| Total effect (path *c*) | 0.080 | 0.038 | 0.034 | – | – |
| Direct effect (path *c'*) | 0.079 | 0.038 | 0.037 | – | – |
| Indirect effect (via mediator) | 0.001 | 0.878 | 0.380 | 0.001 to 0.005 | 0.015 |

Analyses are adjusted for age, sex, ethnicity, wealth, body mass index and status on the outcome variable at baseline (Wave 5).
*P values shown for indirect effects are derived from the Sobel test for consistency with total and direct effects, however bootstrap 95% CIs provide a more robust indication of significant mediation (see Method for more details).
BMI, body mass index; Coeff., coefficient.

diet, dieting behaviour, weight cycling and disordered eating (such as binge eating disorder) to understand how these factors are involved in the relationship between weight discrimination and health. This is of relevance in both community settings (as in this study) and in healthcare settings. There is evidence that healthcare providers can hold negative attitudes and believe stereotypes about people with overweight/obesity.[51 52] This can have negative implications for both the quality of care received and can result in people avoiding healthcare settings for fear of encountering weight discrimination.[52] This is of relevance to the current study as we observed a cross-sectional association between perceived weight discrimination and reports of limiting long-standing illness. Indeed, people with obesity-related health conditions (such as diabetes or heart disease) have more frequent encounters with healthcare professionals than those without these conditions. The exploration of healthcare settings as a location where weight discrimination could be encountered represents an important area for future work.

Unmeasured variables include negative emotions such as psychological distress and feelings of anxiety which could be associated with weight discrimination. It is possible that these emotions could mediate the link between perceived weight discrimination and health, or these could be viewed as negative outcomes in their own right. Social isolation and loneliness play a role in weight-related conditions.[53 54] Changes in social contact or relationships could occur as a result of perceiving weight discrimination. The relationship between weight discrimination, indicators of social connectedness and health remain to be teased out. We included self-rated

health as an outcome in this study, as this is considered to better and more broadly capture 'health' than most other indicators.[55] However, the association between perceived weight discrimination and specific health conditions (including weight-related conditions) represents an important area for future work. Another explanation for the non-significant mediation observed in the current study is that the variables we analysed as potential mediators were not adequately captured by the available measures. For example, hair cortisol was examined in 2 cm long hair samples, and thus reflected approximately 2 months' exposure to circulating cortisol. While there are advantages in using hair cortisol over other more transient cortisol measures (eg, in blood or saliva) 2 months is still a relatively short interval given that these data were collected on average 2 years after discrimination was assessed. In addition, research using hair cortisol is in its infancy and it is unclear whether a 1 SD increase in hair cortisol is clinically meaningful.[56] Further, cortisol does not just reflect stress-related activation of the HPA axis. Cortisol parameters are altered in weight-related conditions such as subclinical hypercortisolism and diabetes.[57 58] Cortisol dynamics are complex, with heightened as well as blunted responses associated with weight-related ill-health.[59 60] These complexities warrant attention in future work. It is also possible that other stress-related biological pathways (such as chronic inflammation) could play a role in the weight discrimination-health relationship. Physical activity was self-reported in response to three basic questions on the frequency of light, moderate or vigorous exercise. Comparisons of self-reported versus objective measures of physical activity

have shown that there are often substantial differences between what people say and what they actually do.[61] Our measures of smoking behaviour and alcohol consumption were crude, as both factors were self-reported using a single item measure. Further research should aim to replicate our analyses with different and more detailed measures of the constructs we have examined (stress, physical inactivity, smoking and alcohol consumption) to establish whether the null results of our mediation analyses are the result of limitations in our measures (such as self-reported health risk behaviours) or an accurate representation of the causal pathways linking perceived weight discrimination with poorer health and well-being outcomes.

Strengths of the study include the large sample, prospective study design and assessment of multiple mediators and outcomes. However, there were several limitations. Weight was not measured in the same data collection wave as discrimination, and participants may have changed weight status since experiencing discrimination. The prevalence of weight discrimination in the sample was low which limits the generalisability of our findings. Further not all participants who perceived weight discrimination provided hair cortisol samples meaning we may not have been able to detect small effects. We did not have complete data on weight, and it is possible that people most troubled by their weight were more likely to decline to be weighed. It is also possible that individuals most vulnerable to the negative effects of weight discrimination opted not to complete the item on weight discrimination. The discrimination questions asked about five broad discriminatory situations and were not tailored for weight discrimination. This may have helped avoid bias or priming, as participants were able to attribute multiple reasons for their experience of discrimination.[31–34 47] However, other measures with more specific items on experiences weight discrimination, for example, having to pay more on public transport for occupying two passenger seats or being viewed unfavourably as a potential romantic partner, may have garnered different results. Our measure was not validated to assess weight discrimination. The use of tools such as the Stigmatising Situations Inventory[62] or the Fat Phobia Scale[63] that include specific items on weight discrimination, weight bias internalisation and the experience of weight discrimination by close family, romantic partners or healthcare professionals may also have produced different findings. Further, weight discrimination was only measured on one occasion meaning we had insufficient information on the dose or duration or this form of discrimination. This increases the risk of misclassification bias.

Finally, while the data were drawn from a representative sample, there was missing data and loss to follow-up, meaning the results may not be representative of the target population. In addition, the sample was comprised of predominantly white middle-aged and older adults. The results may not generalise to other samples, in particular, younger samples who may be more likely to report weight discrimination.[64] We did not have data on other negative emotional states such as psychological distress and anxiety which are likely relevant to the links between weight discrimination, health and well-being.

Overall, these results provide further prospective evidence linking weight-based discrimination with poorer health and well-being. However, while perceived weight discrimination was associated with greater stress exposure and physical inactivity, these variables explained only a small part of the association between weight discrimination and poorer health and well-being. This leaves unanswered questions about the pathways through which weight discrimination adversely impacts health and well-being. Further research is required using large prospective-based studies that have been specifically designed to assess these questions and to test unmeasured factors (eg, diet and eating behaviour). There is a great need to include more diverse samples in this future work to improve the generalisability of study findings and also to understand intersectional issues (eg, how weight discrimination may interact with other forms of discrimination such as racism to influence health). Understanding of mechanisms of action is required if policies and interventions to mitigate the health burden of weight discrimination are to be developed. This is imperative as the literature and the findings of the current study underscore the need to tackle weight discrimination to improve population health.

**Contributors** RAH conducted statistical analyses, wrote, edited and reviewed the manuscript. RAH is the guarantor of this work. SEJ conducted statistical analyses, wrote, edited and reviewed the manuscript. EC edited and reviewed the manuscript. AS edited and reviewed the manuscript.

**Funding** This research was supported by the Economic and Social Research Council (https://esrc.ukri.org/), grant number ES/R005990/1(awarded to AS and SJ), and the Academy of Medical Sciences/the Wellcome Trust/the Government Department of Business, Energy and Industrial Strategy/the British Heart Foundation/Diabetes UK Springboard Award (SBF006\1036) (awarded to RH). The funders had no role in study design, data collection and analysis, decision to publish or preparation of the manuscript.

**Competing interests** None declared.

**Patient and public involvement** Patients and/or the public were not involved in the design, or conduct, or reporting, or dissemination plans of this research.

**Patient consent for publication** Not applicable.

**Ethics approval** Ethical approval for ELSA was obtained from the London Multicentre Research and Ethics Committee (MREC/01/02/91). Participants gave informed consent to participate in the study before taking part.

**Provenance and peer review** Not commissioned; externally peer reviewed.

**Data availability statement** Data are available in a public, open access repository. Data from the English Longitudinal Study of Ageing are available to download from g2aging.org.

terminology, drug names and drug dosages), and is not responsible for any error and/or omissions arising from translation and adaptation or otherwise.

**ORCID iDs**
Ruth A Hackett http://orcid.org/0000-0002-5428-2950
Sarah E Jackson http://orcid.org/0000-0001-5658-6168
Andrew Steptoe http://orcid.org/0000-0001-7808-4943

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
