## [Reviewer comments · BMJ Open]

ARTICLE DETAILS

TITLE (PROVISIONAL)	The role of stress and health behaviour in linking weight discrimination and health: a secondary data analysis in England
AUTHORS	Hackett, Ruth; Jackson, Sarah; Corker, Elizabeth; Steptoe, Andrew

VERSION 1 – REVIEW

REVIEWER	Quatromoni, Paula Boston University, Health Sciences
REVIEW RETURNED	25-Feb-2023

GENERAL COMMENTS	This paper presents a secondary data analysis from the English Longitudinal Study of Ageing in the U.K. In a cohort of 4,341 adults with overweight or obesity, researchers tested the association between perceived weight discrimination and several health outcomes (self-rated health, limiting long-standing illness, depressive symptoms, quality of life, and life satisfaction) both crosssectionally and prospectively four years later. Stress exposure (measured by hair cortisol levels) and a health-risk behaviors of smoking, physical inactivity, and alcohol consumption were evaluated as potential mediators of any observed associations. In prospective analyses, weight discrimination was associated with higher odds of fair/poor self-rated health status and depressive symptoms. While those who reported discrimination had higher stress exposure (hair cortisol levels) and higher odds of physical inactivity, these factors (and the others considered) were not discerned to be mediators of the observed association between weight discrimination and health outcomes. The authors conclude that weight discrimination is, indeed, a public health problem that needs to be addressed. However, the explored variables explain very little of the association observed in this study population, meaning that much more research is needed on this topic. This invites opportunity for the Discussion section to put forth a bit more critical thinking than was presented in this initial version of the manuscript. There were several strengths of this manuscript including the clarity of the writing style, organized presentation of information in the tables and figure, objective SES data and BMI measurement, high return rates of self-completed questionnaires, careful multivariate analysis with a novel mediation
---

	analysis, and inclusion of sensitivity analyses. The authors called out a few limitations and indicated directions for future research. These are two areas that could be expanded upon in the Discussion section where a few missed opportunities are left unattended to, in the opinion of this reviewer. Specific Comments: Title/Abstract – It would be my preference to indicate in the title and/or in the abstract that this paper constitutes a secondary data analysis of a prospective observational cohort study. It feels a bit misleading to present it (in the title and abstract) as a prospective study, as if the study were designed to test these specific hypotheses. Because this is a secondary analysis of an existing data set, several of the methodologic limitations only become known in that context later in the Methods and in the Discussion. As a reader, more transparency is appreciated to set the context for taking in the methods, results and discussion that follow. Strengths and Limitations (page 3) – Please introduce some epidemiologic terms that are salient limitations in this study, like measurement error, misclassification, risk of bias, and limited generalizability. As well, this was a low-risk population in which these hypotheses were tested with fewer than 5% of the cohort “exposed” to weight discrimination. Finally, the secondary data analysis situation precludes investigation of “other” health-risk behaviors that may be really important, but your study did not have data on several of these factors, namely dietary patterns, diet quality, dieting behavior, disordered eating, binge eating, or weight cycling. Feb 2023 Introduction (page 4, lines 39 and 42) – Can you expand a bit on the “negative emotional state” that contributes to the allostatic load and poor health outcomes? Isn’t it more than depression? The mention of “coping” in line 36 in this same paragraph makes me wonder about anxiety and other altered mood states. I wonder why anxiety was not a health outcome in your study (did you not have data on anxiety to consider it?). Another factor at play here could be eating disorders, specifically binge eating disorder (BED), which is a maladaptive coping mechanism in response to stress, including stress from weight-based discrimination. This pattern of what is described as a “vicious cycle of stress, weight gain, and further discrimination” is alluded to in line 53 (page 4), but the concept of an eating disorder or BED specifically is not introduced. Of the factors mentioned (line 55, page 4) as “coping mechanisms” (food, nicotine, and alcohol) to alleviate the negative psychological impact of discrimination, only smoking and alcohol are considered in this paper. While it is mentioned in the Discussion (page 16, line 23) that this study had no information on dietary factors, the lack of direct commentary about disordered eating patterns and binge eating
--	--

	disorder is an important omission, particularly given that the very next sentence makes reference to “overeating and other unhealthy eating behaviors.” It is essential that we name these behaviors and these health concerns. It would be appropriate to describe the potential influence of (and need to study) unsupervised and/or unsafe dieting practices, binge eating, disordered eating, and weight cycling. These are all potential and unexamined factors that may lie in the pathway from weight discrimination to poor health. I believe this is highly relevant, in general and in this specific study. BED is the most common eating disorder, affecting 2-4% of the population (including in the UK); it affects almost equal proportions of men and women; and it quite often affects individuals living in larger bodies (ie. your exact study sample) who are chronically dieting, trying to lose weight, and may be subjected to a disproportionate burden of weight-based discrimination, in general and specifically in healthcare settings. The occurrence of BED in your study population, restricted to the overweight/obese portion of the cohort representing your base population, could be quite high. Yet, binge eating behavior, BED risk (which can be screened for), or BED diagnosis are not measured or reported on your cohort and therefore, not considered in this paper. Given that the factors that you did consider turned out to explain so little of the association between weight-based discrimination and poor health outcomes, it seems important, particularly in the Discussion, for the authors to think critically about “what didn’t we measure that would be important for future studies to investigate?” This would inform a directive for newly designed prospective studies to set out to measure health-related behaviors like dietary patterns, diet quality, dieting behavior, weight cycling, and disordered eating behaviors including binge eating and BED. Another possible contributing factor that has been discussed and documented elsewhere is healthcare avoidance that results from weight-based discrimination that occurs in healthcare interactions with medical providers. The lack of mention or attention to these factors in this manuscript feels like a glaring missed opportunity that I would like to see addressed in the Discussion section, as there is literature to support these assertions. Methods: Exposure: perceived weight discrimination (page 6) – All of the citations in this section are tied back to prior studies by this same author group, applying the same methods of exposure measurement, classification, and interpretation. All appear to be descriptive or hypothesis-testing analysis, not validation studies. Has this tool been validated as a measure of weight-based discrimination? Are there others tools for measuring this exposure that are validated (that maybe should be used in other, newly
--	---

	designed prospective studies)? In this study, we do not have sufficient information on the dose or duration of the weight-based discrimination to know if it constitutes a valid representation of exposure that would in any way relate to the outcomes of interest. This deserves disclosure, at a minimum in the Discussion/Limitations section, given the risk of exposure measurement error and exposure misclassification which would have bearing on the results. Results: Tables 3 and 4 – Please rename the variable as Physical Inactivity since that is the “exposed” group that experienced more weight discrimination (Table 3) and worse health outcomes (Table 4). Page 12, line 18 – Is a one-unit increase in hair cortisol concentration clinically meaningful? Some interpretation of this here or in the Discussion seems relevant. Discussion: Page 15, lines 50+ - Could the failure to replicate a prospective association with long-standing illness somehow relate to misclassification errors in your study, or to more of a bi-directional association with this specific health outcome? It is interesting that this relationship was strong and significant in the cross-sectional context where weight discrimination in the past year was related to the prevalence of longstanding illness. Might this imply that healthcare-related weight-based discrimination is an important source of exposure that may be experienced to a heightened degree and/or more frequently by those with chronic health conditions, especially obesity-related health conditions like type 2 diabetes, CVD, hypertension, knee osteoarthritis, etc. If the longitudinal analysis evaluated “prevalence” but not “new onset” (incidence) of chronic health conditions, there is some misclassification in the outcome. But there is also potential misclassification in the exposure since updated episodes of weight discrimination over time are not taken into account (this variable was only measured at baseline). If the exposure misclassification is random, it would attenuate any observed result towards the null (which is seen in your analysis). If the exposure misclassification is more systematic (ie. weight discrimination is happening to a greater extent in folks who are gaining weight over time and/or who are getting new chronic disease diagnosis during follow-up), then it could introduce bias. This deserves critical thought. Page 15, line 55 – Sample characteristics other than age range deserve mention, like the lack of diversity in race/ethnicity in your study population; and the differences in “covariates used” is a potential problem of uncontrolled confounding. It is helpful to name the epidemiologic concerns for the reader. Page 16, middle paragraph – As noted prior, the discussion falls short here by not diving a bit more into the unmeasured variables. “Dietary factors” are briefly mentioned (line 23) but deserve more thought and commentary as discussed above. What is meant by “psychological distress” in line 30? Do you mean
--	---

	anxiety? Is it a mediating variable (as alluded to here), or is it a potential health outcome like depression? What about healthcare avoidance which is described in the literature as a known consequence of weight discrimination and a known contributor to adverse health outcomes including delayed diagnosis of chronic health conditions and/or inadequate treatment? Page 16, bottom of the paragraph – “Limitations in our measures” are mentioned here in relation to some observed null results. This theme should be better discussed here and later in the next paragraph Feb 2023 devoted to Limitations. Not only was physical activity self-reported, it (and smoking and alcohol exposures) were assessed using relatively crude methods, often a single question with a gross classification scheme. The uncertainty of validity of the exposure measure, the potential misclassification error in the weight discrimination variable, and the overall low occurrence of the exposure in this study sample are other potential contributors to the null results observed. Other methodologic limitations inherent in a secondary data analysis, and those called out above using key epidemiologic terms, would be relevant to include on page 17. Page 17, line 37 – I would like to see the “further research” sentence built out a bit. This paper begs a call to action for new studies designed specifically to study weight-based discrimination and health outcomes. The secondary data analysis presented here is just a starting point. We need larger prospective studies placed in higher risk, more diverse study populations (where racism may interact with weight-based discrimination to have a unique impact), conducted with more methodologic rigor and internal validity, assessing the very important “unmeasured factors” that are not traditionally measured (or already existing) in other data sets (like binge eating behavior and other aspects of eating and dieting behavior) that may mediate the relationship between weight-based discrimination and poor health outcomes. Page 17, line 41 - I recommend restructuring this sentence. The “if not for moral reasons” phrase covertly implies that the moral, ethical reasons for addressing weight-based discrimination are not enough, in and of themselves.
--	--

REVIEWER	Robinson, Katie The University of Iowa
REVIEW RETURNED	03-Apr-2023

GENERAL COMMENTS	 - Overall excellent study. Important topic, well-designed, with appropriate statistical analyses and sensitivity analyses. Although it is essentially a negative study, still very important for the field. - It would be helpful to understand how many participants who perceived weight discrimination endorsed other forms of discrimination (e.g. race or gender). Were there any intersectional effects? Is this a potential confounder?
---

	 - Would be beneficial to comment on why self-reported health status was used rather than a comorbidity index, allostatic load or another measure. Was more specific health data available (e.g. on specific medical conditions such as hypertension or diabetes)? - It would be helpful to comment on the finding of more perceived weight discrimination in less wealthy groups. To my knowledge this is an unusual finding. - Please comment on what effect sizes you are able to detect with your study. Although the sample was large, the number of people reporting perceived weight discrimination and with hair cortisol levels was somewhat low – would you be able to detect small differences? - Limitations: Please comment on cortisol levels as a reflection of many different disease states, not simply stress-related activation of the HPA. Disease states such as subclinical hypercortisolism (which could increase weight and depression) would not necessarily be detected on self-report measures. Cortisol reactivity is also potentially blunted with age, another important consideration. Minor points:  - Is this work particularly important in the elderly? Seems more relevant in younger populations (page 5, line 28). - A sensitivity analysis in higher BMI ranges (e.g. 30+ or 35+ kg/m²) might have been helpful given that weight discrimination generally increase substantially at higher BMIs and mechanisms might differ. - Limitations: could also include there is no measure of weight bias internalization. - Is there a rationale for dichotomizing the 5th item on the perceived weight discrimination scale differently than the other items? - Excellent/ thoughtful discussion of potential additional mediators for perceived weight discrimination and outcomes – could also consider social isolation as a relevant factor in older adults (potentially influenced by perceived discrimination).
--	--

VERSION 1 – AUTHOR RESPONSE

Reviewer 1: Dr. Paula Quatromoni, Boston University

Overall Comments: This paper presents a secondary data analysis from the English Longitudinal Study of Ageing in the U.K. In a cohort of 4,341 adults with overweight or obesity, researchers tested the association between perceived weight discrimination and several health outcomes (self-rated health, limiting long-standing illness, depressive symptoms, quality of life, and life satisfaction) both cross-sectionally and prospectively four years later. Stress exposure (measured by hair cortisol levels) and a health-risk behaviors of smoking, physical inactivity, and alcohol consumption were evaluated as potential mediators of any observed associations. In prospective analyses, weight discrimination was associated with higher odds of fair/poor self-rated health status and depressive symptoms. While those who reported discrimination had higher stress exposure (hair cortisol levels) and higher odds of physical inactivity, these factors (and the others considered) were not discerned to be mediators of the observed association between weight discrimination and health outcomes. The authors conclude that weight discrimination is, indeed, a public health problem that needs to be addressed. However, the explored variables explain very little of the association observed in this study population, meaning that much more research is needed on this topic. This invites opportunity for the Discussion section to put forth a bit more critical thinking than was presented in this initial version of the manuscript.

There were several strengths of this manuscript including the clarity of the writing style, organized presentation of information in the tables and figure, objective SES data and BMI measurement, high return rates of self-completed questionnaires, careful multivariate analysis with a novel mediation analysis, and inclusion of sensitivity analyses.

The authors called out a few limitations and indicated directions for future research. These are two areas that could be expanded upon in the Discussion section where a few missed opportunities are left unattended to, in the opinion of this reviewer.

Thank you for taking the time to provide this detailed review. The reviewer has raised valid points. We provide responses to the issues raised below and have amended the Discussion section in line with the requests above. Specifically, more detail on the limitations of the current study, as well as more in-depth discussion of avenues for future research are now provided in the Discussion section.

Specific Comments:

Title/Abstract – It would be my preference to indicate in the title and/or in the abstract that this paper constitutes a secondary data analysis of a prospective observational cohort study. It feels a bit misleading to present it (in the title and abstract) as a prospective study, as if the study were designed to test these specific hypotheses. Because this is a secondary analysis of an existing data set, several of the methodologic limitations only become known in that context later in the Methods and in the Discussion. As a reader more transparency is appreciated to set the context for taking in the methods, results and discussion that follow.

In response to this comment, we have now amended the title of the study to state that the study was a secondary data analysis.

New title: “The role of stress and health behaviour in linking weight discrimination and health: a secondary data analysis in England”.

We have also added this information to the abstract and to the Strengths and Limitations section.

Strengths and Limitations (page 3) – Please introduce some epidemiologic terms that are salient limitations in this study, like measurement error, misclassification, risk of bias, and limited generalizability. As well, this was a low-risk population in which these hypotheses were tested with fewer than 5% of the cohort “exposed” to weight discrimination. Finally, the secondary data analysis situation precludes investigation of “other” health-risk behaviors that may be really important, but your study did not have data on several of these factors, namely dietary patterns, diet quality, dieting behavior, disordered eating, binge eating, or weight cycling.

In response to this comment, we now include the terms “generalisability”, “misclassification bias” and “measurement error” in this section. We also include two new bullet points acknowledging that 1) weight discrimination was only measured on one occasion and 2) that we were lacking important information on diet and eating behaviour. We hope these additions address the reviewer’s concerns.

Introduction (page 4, lines 39 and 42) Can you expand a bit of the “negative emotional state” that contributes to the allostatic load and poor health outcomes. Isn’t it more than depression?

Thank you for this comment. We now clarify that the “increased negative emotion” in response to perceiving discrimination could include “psychological distress, depressive symptoms, and fear, among others”.

We agree with the reviewer that the “negative emotional state” in response to repeated discrimination goes beyond depression. We now specify that this could include “depression, anxiety, ongoing feelings of stress etc.” (see page 4).

The mention of “coping” in line 36 in this same paragraph makes me wonder about anxiety and other altered mood states. I wonder why anxiety was not a health outcome in your study (did you not have data on anxiety to consider it?).

Unfortunately, we did not have data on anxiety. We now acknowledge this limitation in the Discussion section (see page 17):

“Unmeasured variables including negative emotions such as psychological distress and feeling of anxiety which could be associated with weight discrimination”.

Another factor at play here could be eating disorders, specifically binge eating disorder (BED), which is a maladaptive coping mechanism in response to stress, including stress from weight-based discrimination. This pattern of what is described as a “vicious cycle of stress, weight gain and further discrimination” is alluded to in line 53 (page 4), but the concept of an eating disorder or BED specifically is not introduced.

Thank you for this interesting point. We now introduce how disordered eating patterns and binge eating disorder could play a role as maladaptive coping mechanisms in response to perceived weight discrimination before the line on “vicious cycle of stress, weight gain and further discrimination” that the reviewer highlights (see pages 4-5):

“Indeed, disordered eating and binge eating disorder (eating large amounts accompanied by feelings of distress) is common at higher weight ranges and has been associated with weight discrimination”.

We also include 2 new citations to back up this claim:

Salvia MG, Ritholz MD, Craigen KLE, Quatromoni PA. Women’s perceptions of weight stigma and experiences of weight-neutral treatment for binge eating disorder: a qualitative study. *eClinicalMedicine*. 2023 Feb 1;56:101811.

Wu YK, Berry DC. Impact of weight stigma on physiological and psychological health outcomes for overweight and obese adults: A systematic review. *J Adv Nurs*. 2018;74(5):1030–42.

Of the factors mentioned (line 55, page 4) as “coping mechanisms” (food, nicotine and alcohol) to alleviate the negative psychological impact of discrimination, only smoking and alcohol are considered in this paper. While it is mentioned in the Discussion (page 16, line 23) that this study had no information on dietary factors, the lack of direct commentary about disordered eating patterns and binge eating disorder is an important omission, particularly given that the very next sentence makes reference to “overeating and other unhealthy eating behaviors”. It is essential that we name these behaviors and these health concerns. It would be appropriate to describe the potential influence of (and need to study) unsupervised and/or unsafe dieting practices, binge eating, disordered eating, and weight cycling. These are all potential and unexamined factors that may lie in the pathway from weight discrimination to poor health.

We are sorry for this omission in our initial submission. In response to this comment, we now include some new information on disordered eating patterns and binge eating disorder before and after the sentence on “overeating and other unhealthy behaviours” that the reviewer has flagged in the Discussion Section (page 16):

We had no information on dietary factors or eating behaviours such as unsupervised and/or unsafe dieting practices, binge eating, disordered eating, and weight cycling. Therefore, we were unable to explore these as potential mediators. Given evidence that people who experience weight discrimination are at increased risk of overeating, disordered eating(48,49) and have a greater risk of binge eating disorder (25) than those who do not report discrimination these present important unexamined pathways from weight discrimination to poor health.

I believe this is highly relevant, in general and in this specific study. BED is the most common eating disorder, affecting 2-4% of the population (including in the UK); it affects almost equal proportions of men and women; and it quite often affects individuals living in larger bodies (i.e., your exact study sample) who are chronically dieting, trying to lose weight, and may be subjected to a disproportionate burden of weight-based discrimination, in general and specifically in healthcare settings. The occurrence of BED in your study population, restricted to the overweight/obese portion of the cohort representing your base population, could be quite high. Yet, binge eating behavior, BED risk (which can be screened for), or BED diagnosis are not measured or reported on your cohort and therefore, not considered in this paper. Given that the factors that you did consider turned out to explain so little of the association between weight-based discrimination and poor health outcomes, it seems important, particularly in the Discussion, for the authors to think critically about “what didn’t we measure that would be important for future studies to investigate?”. This would inform a directive for newly designed prospective studies to set out to measure health-related behaviors like dietary patterns, diet quality, dieting behavior, weight cycling, and disordered eating behaviors including binge eating and BED.

We agree with the reviewer that binge eating disorder, diet (quality and pattern), dieting and weight cycling represent important avenues for future work. In response to this comment, we specifically highlight the importance of studying these factors in future work. We highlight the importance of studying these factors in both community and healthcare settings (see pages 16-17).

It is plausible that the use of maladaptive dietary and eating behaviours as a means of coping with weight discrimination could lead to a vicious cycle of further distress, weight gain and disproportionate burden of weight discrimination. This represents an important avenue of further research, particularly considering the lack of mediation by other factors in the current study. Future prospective studies should explicitly set out to measure the role of diet, dieting behaviour, weight cycling, and disordered eating (such as binge eating disorder) to understand how these factors are involved in the relationship between weight discrimination and health. This is of relevance in both community settings (as in this study) and in healthcare settings.

Another possible contributing factor that has been discussed and documented elsewhere is healthcare avoidance that results from weight-based discrimination that occurs in healthcare interactions with medical providers. The lack of mention or attention to these factors in this manuscript feels like a glaring missed opportunity that I would like to see addressed in the Discussion section, as there is literature to support these assertions.

Thank you for this useful suggestion. We now mention this on page 17 and include 2 new references to literature in the area:

There is evidence that healthcare providers can hold negative attitudes and believe stereotypes about people with overweight/obesity. This can have negative implications for both the quality of care received and can result in people avoiding healthcare settings for fear of encountering weight discrimination. This of relevance to the current study as we observed a cross-sectional association between perceived weight discrimination and reports of limiting longstanding illness. Indeed, people with obesity-related health conditions (such as diabetes or heart disease) have more frequent encounters with healthcare professionals than those without these conditions. The exploration of healthcare settings as a location where weight discrimination could be encountered represents an important area for future work.

Phelan SM, Burgess DJ, Yeazel MW, Hellerstedt WL, Griffin JM, van Ryn M. Impact of weight bias and stigma on quality of care and outcomes for patients with obesity. *Obes Rev.* 2015;16(4):319–26.
Sabin JA, Marini M, Nosek BA. Implicit and Explicit Anti-Fat Bias among a Large Sample of Medical Doctors by BMI, Race/Ethnicity and Gender. *PLOS ONE.* 2012 Nov 7;7(11):e48448.

Methods:

Exposure: perceived weight discrimination (page 6) – All of the citations in this section are tied back to prior studies by this same author group, applying the same methods of exposure measurement, classification, and interpretation. All appear to be descriptive or hypothesis-testing analysis, not validation studies. Has this tool been validated as a measure of weight-based discrimination? Are there other tools for measuring this exposure that are validated (that maybe should be used in other, newly designed prospective studies)?

Thank you for this comment. The reviewer is correct in our initial submission we only refer to previous research conducted by our group looking at weight discrimination in ELSA. We apologise for this narrow focus. These questions on perceived discrimination are based on items developed and used widely in other longitudinal studies such as the Health and Retirement Study (HRS) and the Midlife Development in the United States (MIDUS) survey. We now state this on page 6 and include an additional reference to acknowledge this:

Smith J, Ryan L, Sonnega A, Weir DR. Psychosocial and lifestyle questionnaire 2006–2016: Documentation report core section. Ann Arbor, MI, USA.: The HRS Psychosocial Working Group;; 2017.

We acknowledged in our initial submission (page 18) that the measure was not tailored for weight discrimination (as participants were able to attribute multiple reasons for their experience of discrimination). This may be viewed as advantageous as it could have helped avoiding participant priming or bias. We now state this in the Discussion section (see page 18). We also have provided references to other studies that have used this measure to assess other attributions of discrimination (pages 6 and 18).

We also now mention that this tool has not been validated as a measure of weight discrimination and that other tools (e.g., the Stigmatising Situations Inventory or the Fat Phobia Scale) with more specific items on weight stigma, weight bias internalization and experiences of weight discrimination with close family, romantic partners or healthcare professionals may have garnered different results (page 18).

We include references to these other tools:

Bacon JG, Scheltema KE, Robinson BE. Fat phobia scale revisited: the short form. *Int J Obes Relat Metab Disord J Int Assoc Study Obes.* 2001 Feb;25(2):252–7.

Myers A, Rosen JC. Obesity stigmatization and coping: relation to mental health symptoms, body image, and self-esteem. *Int J Obes Relat Metab Disord J Int Assoc Study Obes.* 1999 Mar;23(3):221–30.

In this study, we do not have sufficient information on the dose or duration of the weight-based discrimination to know if it constitutes a valid representation of exposure that would in any way relate to the outcomes of interest. This deserves disclosure, at a minimum in the Discussion /Limitations section, given the risk of exposure measurement error and exposure misclassification which would have bearing on the results.

This is a valid point. In response to this comment, we now acknowledge this limitation in the Discussion section (see page 19).

“Weight discrimination was only measured on one occasion meaning we had insufficient information of the dose or duration or this form of discrimination. This increases the risk of misclassification bias”.

Results:

Tables 3 and 4 – Please rename the variable as Physical Inactivity since that is the exposed group that experienced more weight discrimination (Table 3) and worse health outcomes (Table 4).

We have renamed this variable in the tables in response to this comment.

Page 12, line 18 Is a one-unit increase in hair cortisol concentration clinically meaningful? Some interpretation of this here or in the Discussion seems relevant.

The reviewer raises a valid point. In response to this comment on page 17 we now state:

“In addition, research using hair cortisol is in its infancy and it is unclear whether 1 standard deviation increase in hair cortisol is clinically meaningful”.

We have also included a new reference to back up this point:

Iob E, Steptoe A. Cardiovascular Disease and Hair Cortisol: a Novel Biomarker of Chronic Stress. *Curr Cardiol Rep.* 2019 Aug 30;21(10):116.

Discussion:

Page 15, lines 50+ - Could the failure to replicate a prospective association with long-standing illness somehow relate to misclassification errors in your study, or to more of a bi-directional association with this specific health outcome? It is interesting that this relationship was strong and significant in the cross-sectional context where weight discrimination in the past year was related to the prevalence of longstanding illness. Might this imply that healthcare-related weight-based discrimination is an important source of exposure that may be experienced to a heightened degree and/or more frequently by those with chronic health conditions, especially obesity-related health conditions like type 2 diabetes, CVD, hypertension, osteoarthritis, etc.,

Thank you for this interesting comment. It is possible that healthcare-related weight -discrimination could play a role in the cross-sectional association observed between long-standing illness and reports of weight discrimination in the current study. We now state this on page 17.

This of relevance to the current study as we observed a cross-sectional association between perceived weight discrimination and reports of longstanding illness. Indeed, people with obesity-related health conditions (such as diabetes or heart disease) have more frequent encounters with healthcare professionals than those without these conditions. The exploration of healthcare settings as a location where weight discrimination could be encountered represents an important area for future work.

If the longitudinal analysis evaluated “prevalence” but not “new onset” (incidence) of chronic health conditions there is some misclassification in the outcome. But there is also potential misclassification in the exposure since updated episodes of weight discrimination over time are not taken into account (this variable was only measured at baseline). If the exposure misclassification is random, it would attenuate any observed result towards the null (which is seen in your analysis). If the exposure misclassification is more systematic (i.e., weight discrimination is happening to a greater extent in folks who are gaining weight over time and/or who are getting new chronic disease diagnosis during follow-up), then it could introduce bias. This deserves critical thought.

We did not investigate the association between weight discrimination and specific chronic illnesses in the current study (we now mention this on page 17). We investigated associations between weight discrimination and self-reported longstanding illness that respondents reported as limiting their activities. In our prospective analyses, we took reports of limiting illness at baseline into account. So, the (non-significant) prospective association between weight discrimination and limiting illness is independent of this. Therefore, this association is akin to incidence rather than prevalence. We agree that there could have been misclassification in the exposure of weight discrimination as this was only measured on one occasion. We acknowledge this potential bias on page 18.

“Further, weight discrimination was only measured on one occasion meaning we had insufficient information of the dose or duration of this form of discrimination. This increases the risk of misclassification bias”.

Page 15, line 55 – Sample characteristics other than age range deserve mention, like the lack of diversity in race/ethnicity in your study population; and the differences in “covariates used” is a potential problem of uncontrolled confounding. It is helpful to name the epidemiologic concerns for the reader.

In response to this comment, we now mention the lack of diversity in the current study sample and highlight the potential issue of uncontrolled confounding (see page 16).

Page 16, middle paragraph – As noted prior, the discussion falls short here by not diving a bit more into the unmeasured variables. “Dietary factors” are briefly mentioned (line 23) but deserve more thought and commentary as discussed about.

In response to this (and your earlier comment) we now mention binge eating disorder, diet (quality and pattern), dieting and weight cycling represent important avenues for future work on page 16.

What is meant by “psychological distress” in line 30? Do you mean anxiety? Is it a mediating variable (as alluded to here), or is it a potential health outcome like depression?

We are sorry for the lack of clarity in our initial submission. We now state that psychological distress and anxiety are examples of negative emotions that were unmeasured in our study. The reviewer is correct that these factors could represent mediators through which perceived discrimination could impact health or they could be negative outcomes in their own right. We acknowledge this on page 17.

“Unmeasured variables including negative emotions such as psychological distress and feeling of anxiety which could be associated with weight discrimination. It is possible that these emotions could mediate the link between perceived weight discrimination and health, or these could be viewed as negative outcomes in their own right”.

What about healthcare avoidance which is described in the literature as a known consequence of weight discrimination and a known contributor to adverse health outcomes including delayed diagnosis of chronic health conditions and/or inadequate treatment?

In response to this (and your earlier comment) we now mention the link between weight discrimination and healthcare avoidance on page 17. We also highlight another tool that assesses weight discrimination by healthcare professionals on page 18.

Page 16, bottom of the paragraph “Limitations in our measures” are mentioned here in relation to observed null results. This theme should be better discussed here and later in the next paragraph devoted to Limitations. Not only was physical activity self-reported, it (and smoking and alcohol exposures) were assessed using relatively crude methods, often a single question with a gross classification scheme. The uncertainty of validity of the exposure measure, the potential misclassification error in the weight discrimination variable, and the overall low occurrence of the exposure in this study sample are other potential contributors to the null results observed. Other methodologic limitations inherent in a secondary data analysis, and those called out above using key epidemiologic terms, would be relevant to include on page 17.

Thank you for this comment. In response to this comment, we acknowledge the limitation of our measures of health behaviour and state that this increases the risk of measurement error on page 18:

“Physical activity was self-reported in response to three basic questions on the frequency of light, moderate or vigorous exercise. Comparisons of self-reported versus objective measures of physical activity have shown that there are often substantial differences between what people say and what

they actually do. Our measures of smoking behaviour and alcohol consumption were crude, as both factors were self-reported using a single item measure”.

In addition, we have added more detail on the limitations of our measure of weight discrimination (including the risk of misclassification bias) in the limitations section on page 18.

We acknowledge at the top of page 18 that the prevalence of weight discrimination in the sample was low. In the revised manuscript we explicitly call out that this “limits the generalisability of our findings”.

Page 17, line 37 – I would like to see the “further research” sentence built out a bit. This paper begs a call to action for new studies designed specifically to study weight-based discrimination and health outcomes. The secondary data analysis presented here is just a starting point. We need larger prospective studies placed in higher risk, more diverse study populations (where racism may interact with weight-based discrimination to have a unique impact), conducted with more methodologic rigor and internal validity assessing the very important “unmeasured factors” that are not traditionally measured (or already existing) in other data sets (like binge eating behavior and other aspects of eating and dieting behavior) that may mediate the relationship between weight-based discrimination and poor health outcomes.

Thank you for this helpful suggestion. In response to this comment on pages 18-19 we have expanded this sentence on further research to state:

“Further research is required using large prospective-based studies that have been specifically designed to assess these questions and to test unmeasured factors (e.g., diet and eating behaviour). There is a great need to include more diverse samples in this future work to improve the generalisability of study findings and also to understand intersectional issues (e.g., how weight discrimination may interact with other forms of discrimination such as racism). Understanding of mechanisms of action is required if policies and interventions to mitigate the health burden of weight discrimination are to be developed. This is imperative as the literature and the findings of the current study underscore the need to tackle weight discrimination to improve population health”.

Page 17, line 41 – I recommend restructuring this sentence. The “if not for moral reasons” phrase covertly implies that the moral, ethical reasons for addressing weight-based discrimination are not enough, in and of themselves.

We apologise that this line in our initial submission was inadvertently open to such interpretation. This was not our intention. In response to this comment, we have now removed any reference to the moral or ethical reasons for addressing weight discrimination and instead focus on this as a public health issue. The line now reads (page 19):

“This is imperative as the literature and the findings of the current study underscore the need to tackle weight discrimination to improve population health”.

Reviewer: 2 Dr Katie Robinson, The University of Iowa

Overall excellent study. Important topic, well-designed, with appropriate statistical analyses and sensitivity analyses. Although it is essentially a negative study, still very important for the field.

It would be helpful to understand how many participants who perceived weight discrimination endorsed other forms of discrimination (e.g., race or gender). Were there any intersectional effects? Is this a potential confounder?

Thank you for taking the time to review this paper. A total of 210 participants (4.8% of the sample) reported weight discrimination. In response to this comment, we investigated whether these participants reported other forms of discrimination. A total of 44 (21%) reported sex discrimination, 40

(19%) reported disability discrimination, 11 (5.2%) reported race discrimination and 6 (2.9%) reported sexuality discrimination. We now report these figures on page 9.

It is plausible that these other forms of discrimination play a role in the association between weight discrimination and health and wellbeing. However, this is not something we can meaningfully investigate here due to small cell counts. Further, for race and sexuality discrimination unfortunately few participants in our sample were from ethnic and sexual minority groups.

Therefore, we acknowledge in the concluding paragraph the importance of studying intersectional effects in future:

” To understand intersectional issues (e.g., how weight discrimination may interact with other forms of discrimination such as racism to influence health)”.

Would be beneficial to comment on why self-reported health status was used rather than a comorbidity index, allostatic load or another measure. Was more specific health data available (e.g., on specific medical conditions such as hypertension or diabetes)?

We are sorry it was not clear in our initial submission why we focused on self-rated health over other indicators of health that the reviewer mentions (e.g., comorbidity index). Self-rated health is considered to better and more broadly capture ‘health’ than most other indicators. An example of this is that it is known to predict mortality independent of objective measures of health (e.g., Idler, E. L., & Benyamini, Y. (1997). *Journal of health and Social Behavior*, 21-37). We now state this on page 17 and include a new reference to back this point up:

Jylhä M. What is self-rated health and why does it predict mortality? Towards a unified conceptual model. *Soc Sci Med* 1982. 2009 Aug;69(3):307–16.

The reviewer is correct that there is data on specific health conditions in ELSA. We acknowledge associations between individual health conditions (particularly weight-related conditions) would be interesting to explore in the context of perceived weight discrimination in future research in text on page 17:

We included self-rated health as an outcome in this study, as this is considered to better and more broadly capture ‘health’ than most other indicators. However, the association between perceived weight discrimination and specific health conditions (including weight-related conditions) represents an important area for future work.

It would be helpful to comment on the finding of more perceived weight discrimination in less wealthy groups. To my knowledge this is an unusual finding.

Thank you for this interesting comment. The reviewer is correct the association between weight discrimination and wealth is not consistently observed. Previous research in ELSA detected this relationship, but a comparable association was not found in the Health and Retirement study (Amirova et al., 2022).

In response to this comment, we now state on page 16:

Further, we observed an inverse association between perceived weight discrimination and wealth, suggesting that those of lower socioeconomic position could be more vulnerable to the deleterious effects of weight discrimination. This inverse association is in line with previous reports in ELSA, but not other studies. Therefore, the role of socioeconomic position in the associations between weight discrimination, health and wellbeing, warrants further attention.

We include an additional reference here:

Amirova A, Rimes KA, Hackett RA. Perceived discrimination in middle-aged and older adults: Comparison between England and the United States. *Front Public Health*. 2022.

Please comment on what effect sizes you are able to detect with your study. Although the sample was large, the number of people reporting perceived weight discrimination and with hair cortisol levels was somewhat low – would you be able to detect small differences?

The reviewer is correct, only a small portion of our sample reported weight discrimination (n=210, 4.8%). Of these, 128 had data on hair cortisol. This means we may not have been able to detect small effects. We now state this limitation on page 18 of the manuscript:

The prevalence of weight discrimination in the sample was low which limits the generalisability of our findings. Further not all participants who perceived weight discrimination provided hair cortisol samples meaning we may not have been able to detect small effects.

Limitations: Please comment on cortisol levels as a reflection of many different disease states, not simply stress-related activation of the HPA. Disease states such as subclinical hypercortisolism (which could increase weight and depression) would not necessarily be detected on self-report measures. Cortisol reactivity is also potentially blunted with age, another important consideration. Thank you for raising this relevant point. In response to this comment, we now state on pages 17-18:

Further cortisol does not just reflect stress-related activation of the HPA axis. Cortisol parameters are altered in weight-related conditions such as sub-clinical hypercortisolism and diabetes. Cortisol dynamics are complex, with heightened as well as blunted responses associated with weight-related ill-health. These complexities warrant attention in future work.

We also include some additional references to back up these points:

Chiodini I. Diagnosis and Treatment of Subclinical Hypercortisolism. *J Clin Endocrinol Metab*. 2011 May 1;96(5):1223–36.

Hackett RA, Steptoe A, Kumari M. Association of Diurnal Patterns in Salivary Cortisol With Type 2 Diabetes in the Whitehall II Study. *J Clin Endocrinol Metab*. 2014 Dec 1;99(12):4625–31.

McEwen BS. Stress, adaptation, and disease. Allostasis and allostatic load. *Ann N Y Acad Sci*. 1998 May 1;840:33–44.

Steptoe A, Hackett RA, Lazzarino AI, Bostock S, Marca RL, Carvalho LA, et al. Disruption of multisystem responses to stress in type 2 diabetes: Investigating the dynamics of allostatic load. *Proc Natl Acad Sci*. 2014 Nov 4;111(44):15693–8.

Minor points:

Is this work particularly important in the elderly? Seems more relevant in younger populations (page 5, line 28).

In our initial submission we referenced older populations as we use data from a study of ageing in this manuscript. Also, we mentioned this as older groups are more likely to develop ill-health than younger groups. However, it was not our intention to imply that work on weight discrimination is less relevant in younger populations. We apologise for this. In response to this comment, we have now removed the reference to 'importance' in the introduction paragraph and have re-phrased this as 'interest' (page 4):

"It is of interest to analyse these issues in middle-aged and older people, among whom disease and disability risk is elevated".

A sensitivity analysis in higher BMI ranges (e.g., 30+ or 35+ kg/m²) might have been helpful given that weight discrimination generally increase substantially at higher BMIs and mechanisms might differ.

Thank you for this suggestion. In response to this comment, we now include a sensitivity analysis looking at associations in participants with BMI >30kg/m². As can be seen Supplementary Tables 3 and 4 the pattern of response remains the same at higher BMI levels. We mention this analysis in the revised manuscript on pages 9 and 15.

Limitations: could also include there is no measure of weight bias internalization.

In response to this comment, we now mention on page 18 that tools that assess weight bias internalization could have garnered different results:

The use of tools such as the Stigmatizing Situations Inventory or the Fat Phobia Scale that including specific items on weight discrimination, weight bias internalization and experience of weight discrimination by close family, romantic partners or healthcare professionals may also have produced different findings.

Is there a rationale for dichotomizing the 5th item on the perceived weight discrimination scale differently than the other items?

We are sorry this was unclear in our initial submission. As in previous work, we treated this item differently because responses for this item were skewed with most participants never reporting discrimination in this setting. We now state this on page 6 and provide additional references to other work where this item was treated this way.

Excellent/ thoughtful discussion of potential additional mediators for perceived weight discrimination and outcomes – could also consider social isolation as a relevant factor in older adults (potentially influenced by perceived discrimination).

Thank you for this kind remark and for the interesting suggestion of investigating social isolation in relation to weight discrimination. In response to this comment, we now mention this on page 17 of the manuscript:

Social isolation and loneliness play a role in weight-related conditions. Changes in social contact or relationships could occur as a result of perceiving weight discrimination. The relationship between weight discrimination, indicators of social connectedness and health remain to be teased out.

We also include some additional references to back up these points:

Hackett RA, Hudson JL, Chilcot J. Loneliness and type 2 diabetes incidence: findings from the English Longitudinal Study of Ageing. *Diabetologia*. 2020 Nov 1;63(11):2329–38.

Valtorta NK, Kanaan M, Gilbody S, Ronzi S, Hanratty B. Loneliness and social isolation as risk factors for coronary heart disease and stroke: systematic review and meta-analysis of longitudinal observational studies. *Heart*. 2016 Jul 1;102(13):1009–16.

VERSION 2 – REVIEW

REVIEWER	Robinson, Katie The University of Iowa
REVIEW RETURNED	12-Jun-2023
GENERAL COMMENTS	Comments for Author:

- Overall excellent revisions. Manuscript now effectively addresses important findings including other forms of stigma encountered by participants, socioeconomic status, and limitations of cortisol use.
 - Manuscript also includes an important third sensitivity analysis, an improved discussion of limitations, and a more detailed discussion of what factors might be mediating the relationship between perceived weight discrimination and health outcomes.
 - I appreciate the availability of measures under study.
 - Paper now also includes an important discussion of eating disorders as a potential mediating factor.
- Minor Points:
- Page 54 line 17 "This is"
 - Page 54 line 28 either remove "could" or change "including" to "include" and "feeling" to "feelings"

Summary of the Study:

The current study examines how perceived weight discrimination is associated with self-rated health and measures of well-being, and how these associations may be mediated by stress experiences (as operationalized by hair cortisol levels) and health-risk behaviours (smoking, physical inactivity, and alcohol consumption).

This is a secondary data analysis of a large prospective observational study conducted in the English Longitudinal Study of Ageing (ELSA) cohort. Data were collected on perceived weight discrimination experiences from Wave 5 (2010/2011), hair cortisol, health-risk behaviours, and BMI from Wave 6 (2012-2013) and perceived health and wellbeing from Waves 5 and 7 (2014/2015). Associations between perceived weight discrimination and covariates were examined using ANOVA and χ^2 tests. For primary analyses, the authors used multivariable regression models. All models were adjusted for age, sex, ethnicity, wealth and BMI. Prospective models were also adjusted for baseline status/score on the outcome variable.

The authors first examined for associations between perceived weight discrimination and perceived health and wellbeing cross-sectionally and prospectively. In cross-sectional analyses, they found a significant association between perceived weight discrimination (dichotomized as a Y/N variable) and self-rated health, presence of limiting long-standing illness, depressive symptoms, quality of life and life satisfaction. In prospective analyses, they found a statistically significant association between perceived weight discrimination and self-rated health and depressive symptoms, but not their other variables (presence of limiting long-standing illness, quality of life and life satisfaction). Next, the authors examined whether the prospective associations between perceived weight discrimination and self-rated health and depressive symptoms were mediated by stress exposure and health-risk behaviours. There was no association between perceived weight discrimination and smoking or alcohol consumption. Physical inactivity was associated with perceived weight discrimination, depressive symptoms and fair/poor self-rated health. Hair cortisol levels were associated with perceived weight discrimination and fair/poor self-rated health. This finding did not reach statistical significance, but given borderline p-value of 0.66 the authors examined this relationship further. These relationships were examined using mediation analyses. None of the mediation analyses reached statistical significance.

	The authors concluded that perceived weight discrimination is cross-sectionally associated with higher odds of fair/poor self-rated health, limiting long-standing illness, depressive symptoms, and lower quality of life and life satisfaction, and prospectively associated with fair/poor self-rated health and depressive symptoms. There was no evidence that hair cortisol levels or physical inactivity mediated these prospective associations. General Comments: This paper addresses an important topic. A substantial body of literature supports an association between experienced weight discrimination and detrimental physiological, psychological and behavioral outcomes, but the mechanisms behind these associations are still undergoing investigation despite promising work in this area.¹⁻³ The study is well-designed, taking advantage of a large, prospective cohort study including data on perceived weight discrimination, behavioral health, physical health and mental health. Statistical analyses are appropriate and sensitivity analyses are thoughtfully conducted. Furthermore, the study is well-written. The current work would benefit from a longer follow-up period to clarify some of the relationships under investigation, and hopefully the authors will pursue such analyses at a future date. If such work is undertaken, it would be helpful to re-measure perceived weight discrimination to better understand whether this exposure is limited or ongoing for study participants as this may have differential effects on outcomes. Finally, internalized weight bias could be an important confounder in the current study and could be addressed further. Overall, this is a valuable addition to the literature on perceived weight discrimination and health, and a welcome addition to the field.
--	--

VERSION 2 – AUTHOR RESPONSE

Reviewer 2

1. Page 54 line 17 “This is”

Thank you for spotting is error. This now reads:

“This is of relevance” (see page 17).

2. Page 54 line 28 either remove “could” or change “including” to “include” and “feeling” to “feelings”.

In response to this comment, we have amended the sentence, so it now reads:

“Unmeasured variables include negative emotions such as psychological distress and feelings of anxiety which could be associated with weight discrimination”.